# Review of the Isolation, Characterization, Biological Function, and Multifarious Therapeutic Approaches of Exosomes

**DOI:** 10.3390/cells8040307

**Published:** 2019-04-03

**Authors:** Sangiliyandi Gurunathan, Min-Hee Kang, Muniyandi Jeyaraj, Muhammad Qasim, Jin-Hoi Kim

**Affiliations:** Department of Stem Cell and Regenerative Biotechnology, Konkuk University, 1 Hwayang-Dong, Gwangin-gu, Seoul 05029, Korea; pocachippo@gmail.com (M.-H.K.); muniyandij@yahoo.com (M.J.); qasimattock@gmail.com (M.Q.)

**Keywords:** biogenesis, exosomes, microvesicles, apoptotic bodies, biological functions, therapeutic applications, technical challenges

## Abstract

Exosomes are extracellular vesicles that contain a specific composition of proteins, lipids, RNA, and DNA. They are derived from endocytic membranes and can transfer signals to recipient cells, thus mediating a novel mechanism of cell-to-cell communication. They are also thought to be involved in cellular waste disposal. Exosomes play significant roles in various biological functions, including the transfer of biomolecules such as RNA, proteins, enzymes, and lipids and the regulation of numerous physiological and pathological processes in various diseases. Because of these properties, they are considered to be promising biomarkers for the diagnosis and prognosis of various diseases and may contribute to the development of minimally invasive diagnostics and next generation therapies. The biocompatible nature of exosomes could enhance the stability and efficacy of imaging probes and therapeutics. Due to their potential use in clinical applications, exosomes have attracted much research attention on their roles in health and disease. To explore the use of exosomes in the biomedical arena, it is essential that the basic molecular mechanisms behind the transport and function of these vesicles are well-understood. Herein, we discuss the history, biogenesis, release, isolation, characterization, and biological functions of exosomes, as well as the factors influencing their biogenesis and their technical and biological challenges. We conclude this review with a discussion on the future perspectives of exosomes.

## 1. Introduction

Intercellular communication is necessary to maintain cellular functions and tissue homeostasis in multicellular organisms. This intercellular communication is mediated through direct cell-cell contact or through the transfer of secreted molecules. In order to perform normal functions, cells release various types of vesicles, such as exosomes and microvesicles (MVs) derived from endosomes and the plasma membrane, respectively [1]. These extracellular vesicles (EVs) play a significant role in intercellular communication by serving as a carrier for the transfer of membrane and cytosolic proteins, lipids, and RNA between cells [1]. Other type of vesicles, such as MVs, ectosomes, shedding vesicles, and microparticles are also involved in intercellular communication [1,2,3,4,5]. The release of exosomes occurs by the fusion of multivesicular bodies (MVBs) with the plasma membrane in various types of cells [6,7]. Exosomes have also been isolated from a variety of body fluids, such as blood, semen, saliva, plasma, urine, cerebrospinal fluid, epididymal fluid, amniotic fluid, malignant and pleural effusions of ascites, bronchoalveolar lavage fluid, synovial fluid, and breast milk [8,9]. The specific method used to isolate EVs is critical to the success of the isolation and there have been moves to improve and standardize these methods [10]. Once isolated, vesicles can be analyzed by protein staining, immunoblotting, or proteomic techniques. Several conventional methods have been employed to isolate exosomes. These include differential and buoyant density centrifugation, ultrafiltration, size exclusion, precipitation, and immunoaffinity separation [11]. Among these methods, differential and buoyant density centrifugation is the most widely used approach to extract exosomes from cell culture media and body fluids [10]. To overcome the limitations of time and sample volume of the conventional methods, several companies have developed quick, easy, and reliable isolation kits.

Recently, several characterization and validation methods have been developed, for both research and clinical purposes, to analyze exosome purity and to quantify exosomal cargo. These methods include transmission electron microscopy (TEM), scanning electron microscopy (SEM), atomic force microscopy (AFM), nanoparticle tracking analysis (NTA), dynamic light scattering (DLS), resistive pulse sensing, enzyme-linked immunosorbent assay (ELISA), flow cytometry, fluorescence-activated cell sorting (FACS), and microfluidics and electrochemical biosensors [10,12]. Exosomes originating from a variety of different cells share common structural and functional proteins, such as Rab GTPase, SNAREs, annexins, Alix, Tsg101, and tetraspanins [13,14] and the glycosylphosphatidylinositol-anchored molecules, flotillin, cholesterol, sphingomyelin, and hexosylceramides [15,16]. The release of EVs is stimulated by ATP-mediated activation of purinergic receptors [17], thrombin receptor activation [18], and activation by lipopolysaccharides [19]. Lipopolysaccharides play a significant role in the release of MVs. They change the protein composition of dendritic cells [19,20] and increase the intracellular Ca^2+^ concentration, thus triggering the release of EVs from human erythroleukemia cell lines [21] and mast cells [22].

The functions of EVs depend on their ability to interact with recipient cells and to deliver their contents of proteins, lipids, and RNAs to these cells [1]. The specificity of binding to the target cells is governed by adhesion molecules, such as integrins. Similarly, exosomal tetraspanin complexes are also involved in target cell selection in vitro and in vivo [23], possibly by modulating integrins [24]. Exosomes are involved in a variety of functions, such as the eradication of obsolete molecules [25]; antigen presentation [26]; tumor progression, by promoting angiogenesis and tumor cell migration during metastases [27,28]; differentiation of regulatory T lymphocytes or myeloid cells to suppress immune responses [29]; and dissemination of pathogenesis through interaction with recipient cells [30]. The aim of this review is to discuss the current status and advances in our knowledge of the history, biogenesis, isolation, characterization, biological functions, significance, and role of exosomes. Their potential application in various therapeutic approaches, the current state of the field, and future perspectives and challenges regarding their use are also discussed. This article provides a comprehensive review of exosomes that will assist in the development of more innovative strategies and devices for the simple, efficient, time-saving, and cost-effective use of exosomes for therapeutic purposes.

## 2. Biogenesis of Exosomes

The first and most notable study demonstrating the existence of EVs was published in 1946 [31]. In 1977, De Broe [32] described the release of these “membrane fragments” as a ubiquitous feature of viable cells and until the 1980s, these “fragments” were considered as platelet “dust” or cellular debris that directly budded from the plasma membrane [33]. In 1983, two pioneers, namely Harding and Johnstone, contributed the major discovery that transferrin receptors, associated with small 50 nM vesicles, were released from maturing blood reticulocytes into the extracellular space by a process of receptor-mediated endocytosis and recycling. The term, “exosome,” was coined for these EVs by Rose Johnstone. However, this term was also used to refer to other membrane fragments known as “exosome complexes”, that were isolated from biological fluids and ranged in size from 40 to 1000 nm [34]. In 1980, exosomes were considered as cellular waste disposal units that mediated a novel mechanism of cell-to-cell communication. Later, this term was adopted for 40–100-nm vesicles released during reticulocyte differentiation, as a consequence of multivesicular endosome (MVE) fusion with the plasma membrane [35,36]. Finally, a decade later, exosomes were also found to be released by B lymphocytes and dendritic cells through a similar mechanism [37,38].

EVs are now classified into the following three types based on their mechanism of release and size: exosomes (less than 150 nm in diameter), MVs/shedding particles (100~1000 nm), and apoptotic bodies (>1000 nm) (Figure 1). Exosomes are extracellular membrane vesicles that are a subset of extracellular nanosized membrane vesicles, with a diameter ranging from 30 to 150 nm. Their biogenesis occurs via the exocytosis of multivesicular endosomes and they are released into the extracellular environment during the fusion of multivesicular bodies with the plasma membrane [39]. Multivesicular bodies contain intraluminal vesicles that fuse with the plasma membrane of the parent cell, releasing exosomes into the extracellular space [1,40]. Exosomes are carriers of several molecules, such as DNA, RNA, proteins, and lipids, and their contents directly reflect the metabolic state of the cells from which they originate. 

Exome biogenesis consists of three different stages, including the formation of endocytic vesicles by invagination of the plasma membrane; the formation of MVBs by inward budding of the endosomal membrane; and finally, the fusion of MVBs with the plasma membrane and release of the vesicular contents, called exosomes (Figure 2) [41]. Exosomes have a lipid bilayer and a small cytosol devoid of any cellular organelles. Exosomes are released by the fusion of an organelle of the endocytic pathway, the MVB, with the plasma membrane. The biogenesis of exosomes is a tightly regulated process. It begins with the in-budding of endosomes, which in turn forms MVBs that contain intra-luminal vesicles (ILVs) [42,43]. The MVBs then fuse with the cell membrane of various cell types to release ILVs extracellularly as exosomes. The exosomes then serve as messengers, communicating with other cells by the process of vesicular docking and fusion with the aid of soluble N-ethylmaleimide-sensitive factor attachment protein receptor (SNAREs) complexes and the endosomal sorting complex required for transport (ESCRT), which is important machinery for the synthesis of exosomes. ESCRT consists of four different protein complexes, ESCRT-0, -I, -II, and -III, and the associated AAA ATPase Vps4 complex [44]. Depletion of the ESCRT-0 proteins, Hrs and TSG101, and the ESCRT-I protein, STAM1, reduces the secretion of exosomes. Corroborating these findings, the knockdown of ESCRT-III and its associated proteins, CHMP4C, VPS4B, VTA1, and ALIX, increases exosome secretion [44]. Exosome secretion increases in COS cells after transfection with the integral membrane protein, SIMPLE, whereas a mutated form of SIMPLE interferes with the formation of exosomes [45]. Conversely, Stuffers et al. [46] demonstrated that MVB biogenesis can occur without ESCRTs. For example, ILVs form in the absence of key subunits of all four ESCRT complexes, which indicates that exosomes can form by ESCRT-independent mechanisms. Lipids play a major role in vesicle biogenesis and transport processes, such as membrane deformation, fission, and fusion [47]. For example, ceramide plays a critical role in the formation of exosomes. Ceramide is synthesized from sphingomyelin by sphingomyelinase 2 (nSMase2) and the inhibition of this enzyme reduces the exosomal release of the proteolipid protein in Oli-neu cells [48]. Besides ESCRT proteins, lipids and several other proteins, such as lactadherin, platelet-derived growth factor receptors, annexins, flotillins, GTPases, heat shock proteins, and tetraspanins, are also involved in exosome biogenesis [49,50,51]. Gerst and co-workers demonstrated that homologues of the synaptobrevin/VAMP family, v-SNAREs and t-SNAREs, engage in anterograde and retrograde protein-sorting steps between the Golgi apparatus and the plasma membrane [52,53].

Exosomes are composed of functional proteins, mRNA, and microRNA. In particular, they contain proteins from endosomes, the plasma membrane, and the cytosol, as well as some components from the nucleus, mitochondria, endoplasmic reticulum, and Golgi apparatus. The protein content of exosomes depends on the cell type from which they are secreted. They contain various biomarkers, such as TSG101, charged multivesicular body protein 2a (CHMP2A), Ras-related protein Rab-11B (RAB11B), CD63, and CD81 proteins and lipids, including cholesterol, sphingomyelin, ceramide, and phosphatidylserine [1,40,54,55,56,57]. The macromolecular components of exosomes play a significant role in cellular functions and pathological states, such as inflammation, immune responses, angiogenesis, cell death, neurodegenerative diseases, and cancer [58].

## 3. Factors Influencing the Biogenesis of Exosomes

Although appropriate, optimized techniques are a vital factor to stimulate biogenesis and achieve a high yield of exosomes, certain biological factors are essential for EV formation, secretion, and yield. Studies have suggested that exosome yield depends on the cell type. For example, immature dendritic cells produce limited numbers of EVs [51,59], whereas mesenchymal stem cells produce a large number [60,61]. The confluence of cell cultures also plays a critical role in the biogenesis and secretion of exosomes. Confluent cell cultures produce more exosomes than preconfluent cultures, potentially due to cholesterol metabolism [62]. Contact inhibition between cells can reduce EV secretion, since these cells are entering into quiescence and are not actively dividing [63,64]. Various stimulants, such as Ca^2+^ ionophores [65], hypoxia [66,67,68], and cell detachment [69], can induce the secretion of EVs. Culturing cells in the presence or absence of serum influences monolayer formation and the health of cellular structures and in turn affects the biogenesis of exosomes. These effects depend on the availability of fetal calf serum, which regulates cell differentiation [70,71]. The absence of serum exerts stress on the cells, resulting in phenotypic alterations that are reflected in the protein and RNA content of EVs [72]. Cells cultured in serum-free medium have an increased number of EVs during CM production, compared to cells cultured in EV-depleted medium [73]. Meanwhile, the biogenesis and secretion of EVs are induced in SHSY5Y cells growing on serum-free medium with silver nanoparticles Silver nanoparticles induce the secretion of exosomes under low serum conditions (Figure 3).

## 4. Isolation of Exosomes

### 4.1. Ultracentrifugation

The isolation of pure exosomes is critical to understanding their mechanisms of action and for their application in biomedical sciences. Various techniques have been adopted to facilitate the isolation of exosomes (Figure 4). However, since exosomes are very small, their isolation is challenging. Nevertheless, several labs have successfully isolated exosomes using techniques such as ultracentrifugation, ultrafiltration, chromatography, polymer-based precipitation, and affinity capture on antibody-coupled magnetic beads [74]. However, the method used for the isolation and scale-up of exosomes depends on the type of samples being used. Ultracentrifugation is a conventional method suitable for pelleting lipoproteins, extravesicular protein complexes, aggregates, and other contaminants, but it is not suitable for exosome isolation from clinical samples because it is time-consuming, labor-intensive, requires costly instrumentation, and includes multiple overnight centrifugation steps [75]. Gurunathan et al. [53] used ultracentrifugation and density gradient ultracentrifugation to isolate low-density and high-density vesicles, respectively, in yeast. Density gradient centrifugation has been shown to give the purest exosome population when compared to ultracentrifugation and precipitation-based methods [76]. The size difference between cells, subpopulations of EVs, and proteins allows them to be separated and isolated by centrifugation methods. There are two types of preparative ultracentrifugation—differential ultracentrifugation and density gradient ultracentrifugation. Interestingly, the density of their cargo found in the vesicles alters the separation of exosomes and it seems to be difficult. Ultracentrifugation is suitable for large sample volumes, but not for the small volumes of clinical samples. The ultracentrifugation process requires high centrifugal forces, up to 1,000,000× *g*. There are two types of ultracentrifugation—analytical and preparative [77]. Analytical ultracentrifugation is used to investigate the physicochemical properties of particulate materials and the molecular interactions of polymeric materials, whereas preparative ultracentrifugation is used to fractionate biological components, such as viruses, bacteria, subcellular organelles, and EVs. To overcome the limitations of using these techniques, other simpler isolation methods have been developed that are suitable for use with small sample volumes.

### 4.2. Size-Based Filtration, Size-Exclusion Chromatography, and Polymer Precipitation

One of the most important methods for the size-based isolation of exosomes is ultrafiltration (UF). Isolation using this method merely depends on size or molecular weight. Exosomes can be isolated using membrane filters with defined molecular weight or size exclusion limits [78]. UF is rapid, faster than ultracentrifugation, and does not require costly equipment. High-purity preparations of exosomes and EMVs have been achieved using UF and size-exclusion chromatography (SEC) techniques [79]. In this method, exosomes are isolated by fractionation. Although ultrafiltration results in pure vesicles, the disadvantage of UF is that it is difficult to remove contaminating proteins. Size-exclusion chromatography using Sepharose 2B- or CL-4B-packed columns has been successfully employed to fractionate exosomes from bio-fluids [80]. Commercial membrane filters (e.g., polyvinylidene difluoride or polycarbonate, pore size 50~450 nm) can be used to isolate cells and large EVs from biological samples. Filtration methods are often combined with ultracentrifugation, where membranes are used to sieve cells and large EVs, after which the separation of exosomes from proteins is achieved via ultracentrifugation [81].

SEC allows the separation of exosomes from proteins, but not from MVs, protein aggregates, lipoparticles, macromolecules, or particulate matter. EV separation based on size can be achieved by their differential passage through physical barriers, using filters or chromatography columns. Column chromatography allows for the sequential elution of differently sized EV fractions from a single column. However, filter methods are not suitable for the enrichment of exosomes. SEC has also been used in combination with ultracentrifugation to enrich urinary exosomes with a greater yield than the yields obtained solely by ultrafiltration or ultracentrifugation. For instance, exosomes have been isolated by size-exclusion fractionation from mesenchymal stem cells and analysis by TEM showed that they were intact [82].

Precipitation methods have also been used to isolate exosomes by capturing and collecting exosomes of a certain size, between 50 and 150 nm, in ‘‘polymer nets’’ using simple, rapid, low-speed centrifugation on the bench top at 1500× *g*. These exosome precipitation methods are easy to use and do not require any costly or specialized equipment. This method allows easy integration into clinical usage by exploiting existing technologies, and it is scalable for large sample sizes [78]. The polymer used to isolate exosomes needs to be made from innocuous and inert materials that do not elicit immune responses in vitro or in vivo [83]. The disadvantages of this method are that exosomes of different sizes are mixed together and there is no specificity for non-exosomal material, such as protein aggregates, which may co-isolate with the exosomes [74]. Polyethylene glycol (PEG) has also been used to isolate exosomes. This method allows the separation of exosomes from proteins, but not from MVs, protein aggregates, or lipoparticles. The purpose of using water-excluding PEG is to alter the solubility and dispersibility of exosomes, which facilitates their precipitation from biological fluids. 

### 4.3. Immuno-Affinity Purification of Exosomes

Exosome membranes are known to contain large quantities of proteins. Immunoaffinity methods are suitable for isolating exosomes by exploiting the interactions between those proteins (antigens) and their antibodies and specific interactions between receptors and their ligands [77]. This method is particularly useful when proteins expressed on the surface of exosomes lack soluble counterparts. Exosomes can be isolated from cell culture, tissues, and biological fluids, all of which are mixed populations of biological components. To overcome the impurity of exosome preparations, immuno-affinity purification (IP) methods have been used to selectively capture specific exosomes from a complex population, based on certain surface markers. Immuno-affinity methods are rapid, easy, and compatible with routine laboratory equipment. This approach employs magnetic beads covalently coated with streptavidin, which can be coupled in a high-affinity fashion to any biotinylated capture antibody. Anti-CD63, -CD9, and -CD81 antibodies are typically used to isolate exosomes. These antibodies result in the isolation of specific exosomes. Tauro et al. isolated exosomes from colon cancer cells using immunoaffinity capture, which was shown to be more efficient than ultracentrifugation and density gradient isolation methods. This method provides promising results for the detection of exosomes containing specific exosomal markers, such as CD63 or cancer-specific proteins [84,85]. ELISA is an appropriate method for capturing and quantifying exosomes from plasma, serum, and urine using various specific antibodies. The specificity and yield of exosomes isolated by immunoaffinity methods are comparable to yields from ultracentrifugation. To improve this method, submicron-sized magnetic particles have been used for immunoaffinity capture-magneto-immunocapture and shown to yield 10 to 15 times higher quantities of exosomes than ultracentrifugation [86]. Similarly, exosomes isolated using magnetic microbeads coated with anti-CD34 antibodies have shown typical morphology, biological activity, and molecular profiles. These exosomes may be useful for diagnostic and prognostic applications in acute myeloid leukemia (AML) patients [87]. Immunoaffinity methods using magnetic beads have a higher capture efficiency and greater sensitivity than other microplate-based approaches, by virtue of the larger surface area and a near-homogeneous capturing process. In addition, these methods have no volume limitations [77].

### 4.4. Microfluidics-Based Isolation Techniques

New techniques are essential to meet the challenge of providing high-purity exosomes for clinical settings. Conventional methods face many challenges, including a low yield and purity, time intensiveness, a high cost, and difficulties in standardization. Recently, microfluidics-based technologies have become important techniques for the microscale isolation, detection, and analysis of exosomes, using both the physical and biochemical properties of exosomes. This technique utilizes the usual separation determinants, like size, density, and immunoaffinity, but also innovative sorting mechanisms, such as acoustic, electrophoretic, and electromagnetic manipulations; nanowire-based traps (NTs); nano-sized deterministic lateral displacement (nano-DLD); and viscoelastic flow. Furthermore, these methods are quick and consume low volumes of sample and reagents [77,88,89,90]. Immunoaffinity capture on a microfluidic chip enhances specificity and subtyping capability [91]. Wang et al. [92] have developed porous silicon nanowire-on-micropillar structures to differentiate between exosomes and all other EVs and cellular debris. This microfluidic device preferentially traps exosomes with diameters between 40 and 100 nm, while filtering out proteins, other EVs, and cellular debris. Microfluidics-based immunoaffinity capture (Mf-IAC) uses “capture antibodies” or “capture beads” targeting specific surface markers of EV subpopulations [90]. Kanwar et al. [93] efficiently captured circulating EVs using an “ExoChip” platform. Similarly, exocytic vesicles have been efficiently isolated from the plasma membrane using a mica surface with an antibody coating [94]. Shao et al. [95] isolated exocytic vesicles with a high yield, using magnetic capture beads and a magnet-separating Mf-IAC system. 

## 5. Characterization of Exosomes

### 5.1. Nanoparticle Tracking Analysis

Evaluation of the physicochemical properties of exosomes, such as size, shape, surface charge, density, and porosity, is required to determine their biological interactions and hence, the accurate determination of these characteristics is of the utmost importance. Several techniques have been routinely used to characterize exosomes. These include NTA, DLS, resistive pulse sensing, flow cytometry, electron microscopy, and AFM (Figure 4). However, each of these techniques has its own limitations that must be taken into consideration. Exosome characterization has been performed using various methods, including biophysical, molecular, and microfluidic methods. Biophysical methods are used to characterize the exosomal size range. One such biophysical approach is optical particle tracking, such as by NTA, which can measure the concentration and size distribution of exosomes in the 10 nm to 2 µm range. The path of exosomal movement is detected in order to measure the velocity of the particles [12]. This method allows the tracking of the Brownian motion of nanoparticles in a liquid suspension on a particle-by-particle basis. NTA measures the movement of exosomes by tracking each particle through image analysis. This movement can then be correlated to particle size [96]. The outputs of this method are the size, size distribution, concentration, and phenotype of the particles. The advantage of using NTA is its ability to detect different EVs, including exosomes, and to measure small particles with diameters as low as 30 nm. Sample preparation is very quick and easy using this method and the measurement itself only takes minutes. Moreover, samples can be recovered in their native form after the measurements are performed, which makes this technique even more attractive [97]. In addition, the method can detect the presence of antigens on EVs by applying fluorescently labeled antibodies [12]. The crucial parameters for the success of NTA are sample preparation and the correct dilution factor. 

### 5.2. Dynamic Light Scattering

DLS, also known as photon correlation spectroscopy, is an alternative technique for measuring the size of exosomes. The working principle of DLS is that a monochromatic coherent laser beam passes through a suspension of particles [97]. Time-dependent fluctuations in scattering intensity caused by constructive and destructive interference resulting from the relative Brownian movements of the particles within a sample are then observed. This method is simple to use, but it does not visualize the particles. The merit of this method is its ability to measure particles ranging in size from 1 nm to 6 µm. It is most suitable when measuring one type of particle in a suspension (monodispersed suspension). When larger vesicles are present in the suspension, even at a low quantity, the detection of smaller particles becomes problematic [98,99]. The effectiveness of this method has been demonstrated by assessing the distribution and size of EVs in red blood cells [100] and EVs derived from ovarian cancer cells [101]. The technique can provide the diameter range of analyzed vesicles, but it does not provide any biochemical data or information about the cellular origin of EVs [101]. 

### 5.3. Resistive Pulse Sensing

Recently, tunable resistive pulse sensing (TRPS) has emerged as a new technique. It is most useful when measuring the size distribution and concentration of exosomes and when characterizing colloidal particles ranging from approximately 50 nm in diameter up to the size of cells, which is essential when investigating cellular function and uptake [102]. The salient feature of this technique is the in situ single-particle characterization and concentration measurement of exosomes. It is capable of non-subjective characterization on a particle-by-particle basis. TRPS has been used to successfully measure a variety of nanoparticle suspensions, including magnetic beads and a variety of biomolecules. The disadvantage of this technique is that TRPS measurements are susceptible to system stability issues, where the pore can become blocked by particles, and sensitivity issues, where particles are too small to be detected against the background noise of the system. The authors have demonstrated that the system sensitivity and stability can be improved by the optimization of system parameters, such as system noise, sensitivity cutoff limits, and accuracy [102]. Vogel et al. [103] demonstrated the robustness and versatility of TRPS by measuring multimodal mixtures of carboxylated and bare polystyrene particles, mixed anionic and cationic liposomes and exosomes, and by performing an in situ time-course study of DNA attachment onto magnetic nanoparticles. Patko et al. [104] used this technique to analyze leukemia-derived EVs binding to the extracellular matrix, with a size range between 200 and 300 nm. TRPS has been extensively used to measure the size distributions of EVs designed to deliver enzymes to combat Alzheimer’s disease (150–200 nm) and anticancer miRNAs to tumor cells [105,106]. 

### 5.4. Atomic Force Microscopy

AFM is a unique alternative to optical and electron diffraction techniques for studying exosomes. It detects and records interactions between a probing tip and the sample surface and it appears to be a reliable technique. An important feature of this technique is its ability to measure samples in native conditions, with minimal sample preparation and without any destructive mode of operation [107,108]. AFM is used as a nanoscale tool to characterize the abundance, morphology, biomechanics, and biomolecular make-up of exosomes. This technique has contributed to our understanding of exosomes at the single-vesicle and sub-vesicular levels and has also provided useful information regarding the structural, biophysical, and biomolecular characteristics of a variety of sub-cellular structures, such as DNA, membrane proteins, and vesicles. AFM can be used to quantify and simultaneously probe the abundance, structure, biomechanics, and biomolecular content of individual exosomes and other EVs within heterogeneous populations, such as in tumor samples [109]. AFM allows the measurement of the out-of-plane dimensions of nano-objects, with a sub-nanometer accuracy. The disadvantage of these methods is that characterization of the sample was carried out from external analyses, thus under different experimental conditions, like, for instance, temperature, state of the AFM tip, force between probe and sample, or varying scan speed. Several studies have reported the effective use of AFM to characterize EVs derived from blood [108,110], saliva [111], and synovial fluid [5]. Moreover, these studies have described the isolation, detection, membrane composition, mechanical properties, and qualitative analysis of the morphology and size of different types of cell-derived EVs. 

### 5.5. Transmission Electron Microscopy

TEM is a technique that is widely used to characterize the structure, morphology, and size of various biological components. The working principle of TEM is the creation of images as a beam of electrons passes through a sample, where a secondary electron is generated. These electrons are collected and magnified using special lenses. In studies of biological samples, two types of EM are widely used—TEM and cryo-electron microscopy (cryo-EM). Prior to the examination under TEM, specimens need to be fixed in glutaraldehyde and dehydrated. TEM images need to be taken under a vacuum. TEM is used exclusively for the visualization of EVs and the images obtained can then be used for vesicular diameter measurements. An important consideration when using TEM is the sample preparation, which is extensive, involves multiple steps, and may induce changes in the morphology of the EVs. Using TEM, Colombo et al. [40] observed that spherical exosomes and EVs are heterogeneous in shape. Moreover, in some cases, the electron beam may also damage biological samples. The characteristic feature of isolated exosomes examined by TEM is a cup-shaped structure; however, frozen exosomes examined by cryo-TEM show round structures [1]. To avoid the damage caused by the electron beam, cryo-EM is being applied for EV analysis, since it utilizes a different sample preparation protocol. Cryo-EM is free from the effects of dehydration and fixation, because the samples are under liquid nitrogen and under these conditions, the cells are intact, with no ultrastructural changes or redistribution of elements. Cryo-TEM is considered the best method for visualizing nanoparticles and proteins without dehydration artifacts. Cryo-TEM is also suitable for capturing images of exosomes, with membrane structures and lumens. The most important aspect of studying the biological functions of exosomes is the tracing of specific proteins inside exosomes. Generally, specific fluorescent dyes are used to label and visualize specific exosomal proteins. However, in some cases, exosomes, depending on their size and shape, cannot be distinguished using this method because of exaggerated fluorescence signals [112]. Therefore, an alternative method is the visualization of exosomes with specific antibody binding using immunogold EM, which clearly determines the function of these proteins. 

### 5.6. Flow Cytometry

Flow cytometry is a molecular approach used to characterize exosomal surface proteins. It also allows measurement of the size and structure of exosomes [113]. Flow cytometry is one of the most frequently used techniques for EV analysis, as it has the ability to determine the cellular origin of single EVs. The initial sample volume plays a significant role in the isolation and characterization of exosomes using this technique. Therefore, ultracentrifugation remains one of the most reliable techniques, followed by western blotting, NTA [12], and electron microscopy. However, none of these techniques show promise for diagnostic or clinical research applications. By contrast, flow cytometry is a technique that is well-adapted to the reproducible analysis of clinical samples, allowing the analysis of different physical and chemical characteristics of cells and particles in suspension and allowing measurement of the size and structure of exosomes. Conventional flow cytometers can measure particles greater than 300 nm, but are not able to detect smaller particles, based on forward scattered light (FSC). Therefore, these instruments do not allow the direct detection of exosomes [114]. The working principle of a flow cytometer is that a laser beam with a specific wavelength is directed through a stream of fluid containing suspended particles. The degree of light scattering depends on the presence of particles in the samples. Moreover, this technique measures particles labeled with fluorescent dyes. Based on this phenomenon, flow cytometry is able to analyze the relative size and granulation of particles [97], but due to size detection limitations, a significant number of particles escape detection by conventional flow cytometers. Therefore, recently, a high-end, dedicated flow cytometer with higher-sensitivity forward scatter detection, fluorescent amplification, and high-resolution imaging has been developed to distinguish stained exosomes from background contaminants [115,116]. The new generation of flow cytometers use multiple angles for FSC detection, which provides an improved particle resolution [117]. The advantages of this technique are that it allows the rapid measurement of suspended exosomes, it can detect EVs smaller than 300 nm, and exosomes can be quantified and/or classified according to the level of antigen expression [118].

In addition, several other molecular approaches are available to characterize exosomes. These include Raman spectroscopy, which is based on the illumination of samples by a laser. This technique provides information on the chemical structure of exosomes [119]. A microfluidic-based technique has been used to characterize the binding of exosomes to specific antibodies on microfluidics channels and then elute the bound vesicles [120]. Furthermore, exosomes can be characterized by the presence of their cargo molecules, such as RNA. RNA content can be analyzed by microarray analysis, next-generation sequencing, and digital droplet PCR [121]. 

## 6. Biological Functions of Exosomes

Exosomes can be released from a variety of cells including fibroblasts, intestinal epithelial cells, neurons, adipocytes, and tumor cells, and they are found in many biological fluids, such as synovial fluid, breast milk, blood, urine, saliva, amniotic fluid, and malignant effusions of ascites. The biology, function, and heterogeneity of exosomes depend on the cell of origin and the status of the originating tissue or cell at the time of exosome generation. Previous studies have suggested that exosomes may function as cellular garbage bags that expel excess and/or nonfunctional cellular components. Additionally, endocytic vesicles are involved in the recycling of cell surface proteins and signaling molecules [36,122]. Recent studies have shown that exosomes play significant roles in various biological processes, such as angiogenesis, antigen presentation, apoptosis, coagulation, cellular homeostasis, inflammation, and intercellular signaling (Figure 5). These roles are attributed to their ability to transfer RNA, proteins, enzymes, and lipids, thereby affecting physiological and pathological processes in various diseases, including cancer, neurodegenerative diseases, infections, and autoimmune diseases.

### 6.1. Role of Exosomes in Angiogenesis

Angiogenesis is the formation of new capillaries from existing blood vessels and is mediated by a complex multistep process of cellular events [123,124]. Intercellular communication plays a significant role in integrating complex signals in multicellular eukaryotes. For instance, during recirculation and trans-endothelial migration processes, vascular endothelial cells and T lymphocytes closely interact with each other. In support of this hypothesis, Kaur et al. [125] demonstrated that T cell-derived EVs alter endothelial VEGF signaling, tube formation, and gene expression. Interestingly, EVs derived from JinB8 cells enhanced basal VEGFR2 phosphorylation, suggesting that CD47 may indirectly modulate VEGF–VEGFR2 signaling in angiogenesis by targeting endothelial cells via EV trafficking. Tumor exosomes play a prominent role in angiogenesis [126]. Melanoma-derived EVs are able to recruit disseminated melanoma cells and stimulate metastatic factors involved in angiogenesis and extracellular matrix remodeling [28]. The uptake of cancer cell-derived exosomes by endothelial cells stimulates angiogenesis under hypoxic conditions by stimulating the proangiogenic secretome of endothelial cells [68]. Cancer cell-derived exosomes impair the structural integrity of ECs. Cancer cell–derived exosomes loaded with miR-105 downregulate the tight junction protein, ZO-1, thereby enhancing the vascular permeability and metastatic dissemination of endothelial cells [127]. Exosomes loaded with miR-132 from mesenchymal stem cells promote angiogenesis by increasing the tube formation of ECs. Moreover, the subcutaneous injection of human umbilical vein endothelial cells (HUVECs), pretreated with miR-132 exosomes, in nude mice significantly increases their angiogenic capacity in vivo and neovascularization in the peri-infarct zone and preserves heart function [128]. Trophoblast cells shed EVs and extracellular matrix metalloproteinase inducer (EMMPRIN) released in these EVs may regulate angiogenesis, tissue remodeling, and growth in the placenta [129,130]. CD105-positive MVs released from human renal cancer stem cells promote angiogenesis [131]. Progenitor cells release exosomes that stimulate endothelial cell migration [132], cell proliferation [133], tissue vascularization, and angiogenesis [134]. Several miRNAs, such as miR210, miR126, miR132, and miR21, loaded in exosomes from mesenchymal stem/stromal cells (MSCs), play significant roles in angiogenesis [135]. MSC-derived exosomes loaded with multiple miRNAs regulate cell cycle progression and proliferation (miR-191, miR-222, miR-21, let-7a), modulate angiogenesis (miR-222, miR-21, let-7f), and induce EC differentiation (miR-6087) [136]. Exosomes derived from human umbilical cord mesenchymal stem cells enhance angiogenesis during the repair of skin after second-degree burn injury [137]. MSC-derived exosomes significantly promote myogenesis and angiogenesis, compared to MSC-conditioned media [138]. Atienzar-Aroca et al. [139] reported that excessive levels of reactive oxygen species cause the release of large numbers of exosomes from retinal pigment epithelial cells. These exosomes contain high levels of *VEGFR* mRNA expression and can induce angiogenic processes. Cancer stem cells (CSCs) are non-homing, resident tumor stem cells that promote angiogenesis, drug resistance, and metastasis and through local interactions with various cancer cell populations, they regulate tumor growth and progression in multiple cancer types [140,141]. Exosomes derived from CSCs increase the angiogenic potential of endothelial cells [142]. The liver CSCs, CD901, release exosomes that stimulate angiogenesis by upregulating VEGFR1 expression in endothelial cells via long non-coding RNA H19 [143]. Angiogenesis is associated with various physiological processes, including the proliferation, migration, and tube formation of endothelial cells and vascular smooth muscle cells. Their involvement in differentiation, neovascularization, enhanced blood flow restoration, and capillary network formation indicates that exosomes may be a novel therapeutic approach for the treatment of ischemic diseases [144].

### 6.2. Role of Exosomes in Apoptosis

Apoptosis is a highly regulated process that occurs in healthy cells for normal turnover, but also occurs in disease conditions, such as inflammation, infection, autoimmunity, and cancer. In malignant disease, oncogenic mutations cause an imbalance of homeostatic conditions, such as cell viability. Apoptotic cells undergo a series of morphological changes, resulting in the dismantling of the dying cell. Recently, the disassembly of the apoptotic cell has been divided into three distinct morphological steps – blebbing of the apoptotic membrane; formation of a thin membrane protrusion; and ultimately, the generation of apoptotic bodies ranging in size from 1 to 5 µm [145]. Proteomics analysis of exosomes and apoptotic vesicles shows the differential enrichment of proteins between each vesicle type and an increase in the number of vesicles in apoptotic cells [146]. Apoptotic cells in tumors communicate with neighboring cells by intercellular contact and via soluble and EV-encapsulated signal mediators [147]. Apoptotic bodies, a major class of EVs, are released from dying cells as products of apoptotic cell disassembly [148]. Apoptotic bodies display broad size heterogeneity from around 50 nm to several microns [149], and they can be derived from other organelles, such as mitochondria. The critical definition of an apoptotic body is a vesicle that is apoptosis-dependent and encapsulates a wide variety of bioactive molecules and cellular organelles [150]. Conversely, stromal cell-derived EVs (<100 nm), released as a consequence of cell stress, may provide key signals supporting the neighboring tumor cells’ capacity to metastasize, promote proliferation, and inhibit apoptosis [151]. Exosomes derived from liver metastases of colorectal cancer (CRC), carrying miR-375, affect CRC cell apoptosis through the Bcl-2 pathway [152]. Exosomes derived from cancer cells inhibit cell proliferation, have cytotoxic effects on natural killer (NK) cells, and induce T cell apoptosis by carrying the Fas ligand [153,154]. Treatment-induced premature senescence is associated with a significant increase in exosome-like MVs in a p53-dependent manner. Ultrastructural analysis has shown that the RNA interference-mediated knockdown of Tsg101 increases the release of exosomes. These findings suggest that exosomes can transfer cargo with both immunoregulatory potential and genetic information [155]. MVs induce morphological changes, apoptosis, and thrombogenicity in HUVECs; disrupt cellular integrity; and rapidly induce membrane blebbing in ECs [156]. Bruno et al. [157] have shown that BM-MSC-derived exosomes induce apoptosis and cell cycle arrest in HepG2 cells and induce tumor suppression in SCID mice. 

### 6.3. Role of Exosomes in Antigen Presentation

Exosome-mediated signaling induces inflammatory responses by delivering a diverse array of bio-macromolecules, including long and short coding and non-coding RNAs, proteins, and lipids (Figure 5). Exosomes secreted by antigen-presenting cells (APCs) can confer therapeutic benefits by attenuating or stimulating the immune response by carrying and presenting functional major histocompatibility peptide complexes that modulate antigen-specific T cell responses. Dendritic cell (DC)-derived exosomes show immunostimulatory properties by activating T and B cells and exosomes derived from macrophages, and DCs show immunosuppressive properties [158]. Innate and adaptive responses collectively contribute to the overall immune response. The innate immune system is activated by a limited number of receptors that can recognize pathogen-associated or damage-associated molecular patterns [159,160]. The presence of pre-miRNAs and snoRNAs in exosomes induces temporal epigenetic regulation in recipient cells, which regulates the course of inflammatory gene expression [161]. The immune synapse promotes the exchange of miRNA-loaded exosomes between a T cell and its cognate APCs. After stimulation with lipopolysaccharide (LPS), THP-1 cells, a human-derived monocytic cell line, and RAW 264.7, mouse macrophages show differential biomolecular signatures and noncoding RNA populations. Thus, exosome secretion is influenced by the species and the physiological state of the originating cells. Exosomes derived from LPS-stimulated RAW 264.7 cells have increased levels of cytokines and chemokines [162]. EVs released into the immune synapse are responsible for the delivery of cytokines at the synapse. These EVs are derived from plasma membrane-derived MVs or exosomes that are delivered by the polarized fusion of multivesicular endosomes with the plasma membrane at the synapse domain [163,164]. EVs derived from mature and immature DCs show significant differences in the cargo levels of CD86, intercellular adhesion molecule 1, and milk fat globule–epidermal growth factor–factor VIII. These differences are due to both changing protein expression profiles and changing subcellular protein distributions during DC maturation [165].

MHC-I molecules are able to bind and present foreign protein-derived peptides to CD8^+^ T cells. The recognition of such complexes by CD8^+^ cytotoxic T lymphocytes (CTLs) occurs via peptide-specific cognate TCRs. To stimulate the generation of CTLs with specificities toward relevant antigens, cognate naive CD8^+^ T cells need to be activated by DCs. DCs have the unique ability to present peptides derived from exogenous proteins that are acquired by endocytic processes. EVs isolated from activated human monocyte-derived DCs can stimulate peripheral CD8^+^ T cells [166,167]. MHC-II is constitutively expressed only by professional APCs, including DCs, B cells, and macrophages and some epithelial cells. These molecules are transported from the endoplasmic reticulum to endosomes with their peptide-binding groove occupied by the molecular chaperone invariant chain (CD74). DC-derived EVs can induce humoral immune responses that are effective at inducing protective humoral and cellular immune responses against the parasite, *Toxoplasma gondii* [168,169]. 

Macrophages play a significant role in innate and adaptive immune responses by removing pathogens through phagocytosis. Macrophages can also function in antitumor immunity by cross-presenting dead cell-associated antigens to initiate CD8^+^ T cell responses (144). After the phagocytic uptake of various pathogens, such as mycobacteria, salmonella, or toxoplasma, macrophages increase their release of EVs [170,171,172,173,174]. MHC-I and MHC-II expression in epithelial cells hinders their direct contact with T cells; however, MHC-II-carrying EVs can efficiently activate cognate CD4^+^ T cells [175,176,177]. T cell-derived EVs can be targeted to different types of immune cells and can modify their function. EVs derived from activated T cells carry bioactive membrane-associated FasL and the TNF-related apoptosis-inducing ligand (TRAIL). These EVs can prevent potential autoimmune damage by eliminating activated T cells. Activated DCs release MVs and exosomes that can facilitate DC-T cell interactions and have an immunomodulatory role in the intraclonal competition of T cells, respectively [178,179,180]. DC-derived EVs contain costimulatory molecules, as well as relevant tumor-specific antigens loaded onto both MHC-I and MHC-II, which makes them more effective in cancer immunotherapy (Figure 5).

### 6.4. Role of Exosomes in Inflammation

Inflammation is an immune response against infection that is initiated by white blood cells to restore tissue homeostasis. However, uncontrolled or unresolved inflammation can lead to tissue damage, causing several diseases characterized by chronic inflammatory states [181]. Systemic inflammation is an essential component in the pathogenesis of several diseases. Exosomes are known to be involved in inflammatory processes that play a pivotal role in many pathologic states, including cancer, inflammatory bowel diseases, type 2 diabetes, obesity, rheumatoid arthritis, and neurodegenerative diseases [182,183]. The association between inflammation and the levels of specific exosomal cargo molecules is a crucial step in the identification of possible novel biomarkers of inflammatory-based diseases. Exosome-mediated inflammatory responses are involved in all stages of tumor development, immune surveillance, and resistance to therapy [184,185]. Wu et al. [186] showed that exosomes derived from gastric cancer cells could stimulate the NF-κB pathway in macrophages, leading to increased levels of pro-inflammatory factors, which in turn promote tumor cell proliferation and migration. Circulating exosomes induce the secretion of pro-inflammatory cytokines, such as interleukin 6 (IL-6), tumor necrosis factor alpha (TNFα), granulocyte colony-stimulating factor, and chemokine C-C motif ligand 2 (CCL2) [187]. Exosomes that contain miR-21 and miR-29a trigger a pro-metastatic inflammatory response mediated by a Toll-like receptor [188]. Abusamra et al. [189] observed that exosomes derived from human prostate cancer cells inhibit T-cell proliferation and induce T-cell apoptosis through the FasL pathway. Tumor-derived exosomes that are enriched in miRNA(s) influence T cell proliferation and differentiation by downregulating the MARK1 signaling pathway [190]. Inflammatory bowel diseases (IBDs) are a set of chronic disorders that occur due to a failure in maintaining intestinal homeostasis [191]. The introduction of exosomes from IBD patients to the human colonocyte cell line, DLD-1, causes an increase in the level of the pro-inflammatory cytokine, IL-8 [192]. 

EVs promote thrombosis, inflammation, and immune-mediated disease. Exosomes that contain an increased mRNA level of the inflammatory chemokine, *CCL2*, cause inflammation. Injecting mice with CCL2 causes tubulointerstitial inflammation, an enhanced inflammatory response, and macrophage migration, whereas increased levels of CCL2 in humans induce proteinuria in nephropathy patients [193]. Exosomes derived from adipose tissues contain altered levels of 55 miRNAs, most of which regulate TGF-β and Wnt/β-catenin signaling. These signaling pathways regulate the development and progression of chronic inflammation and insulin resistance [194,195]. Synovial fibroblast-derived EVs treated with interleukin-1β induce osteoarthritis-like changes in in vitro and ex vivo models [196]. Neuroinflammation, characterized by a high level of pro-inflammatory cytokine production and glial cell activation, is a common feature of many neurodegenerative diseases [197]. Communication between the cells by exosomes containing α-synuclein, amyloid β, and prions triggers inflammatory signaling in Parkinson’s, Alzheimer’s, and Creutzfeldt-Jacob diseases [198]. All this evidence suggests that exosomes play a significant role in inflammation and immune responses through the modulation of gene expression and cell function (Figure 5).

### 6.5. Exosomes as Biomarkers

Exosomes are released by cells under both normal and pathological conditions. They carry several types of cargo molecules, such as nucleic acids and proteins and are, therefore, considered to be crucial for the discovery of biomarkers for clinical diagnostics (Figure 6). For example, exosomes loaded with tumor-specific RNAs are used as biomarkers for cancer diagnosis and exosomal proteins are considered to be potential biomarkers for a variety of diseases, including cancer, liver disease, and kidney disease. They contain various biomarkers, such as TSG101, charged multivesicular body protein 2a (CHMP2A), Ras-related protein Rab-11B (RAB11B), CD63, and CD81 proteins and lipids, including cholesterol, sphingomyelin, ceramide, and phosphatidylserine [1,40,54,55,56,57]. The macromolecular components of exosomes play a significant role in cellular functions and pathological states, such as inflammation, immune responses, angiogenesis, cell death, neurodegenerative diseases, and cancer [58].

Tetraspanins, a family of membrane scaffold proteins, CD63, and CD81 are also potential biomarkers for cancer. Furthermore, elevated levels of CD81 are associated with inflammation and the severity of fibrosis, indicating that CD81 may be a potential biomarker for hepatitis C diagnosis and treatment response [199,200]. Exosomes isolated from the serum of glioblastoma patients show increased levels of glioblastoma-specific epidermal growth factor receptor (EGFR) vIII, suggesting that it could be used as a diagnostic marker for cancer [201]. Exosomes containing amyloid peptides tau-phosphorylated at Thr-181 are potential biomarkers for Alzheimer’s disease [202,203]. Similarly, exosomes enriched in α-synuclein and the autolysosomal proteins, cathepsin D and the lysosome-associated membrane protein, play a central role in Parkinson’s disease and therefore, these exosomes are also potential biomarkers [51,204]. Acute kidney injury patients have increased levels of fetuin-A and proteins associated with the EGFR pathway, including the alpha subunit of the Gs protein, resistin, and retinoic acid-induced protein 3 in urinary exosomes. Thus, these are potential biomarkers for kidney disease and bladder cancer [205,206]. Serum exosomes enriched in proteoglycan glypican-1 (GP1) are used as a biomarker for pancreatic cancer [207]. All these biomarkers may help avoid surgical procedures for the diagnosis cancers. 

Exosomes not only serve as vehicles to carry RNAs, but also function to protect them from RNase-dependent degradation. Exo-miRs are one type of exosome cargo. Their involvement in various types of cancer-related processes, such as angiogenesis and metastasis, suggests that they can be considered as non-invasive biomarkers for cancer diagnosis [55,208]. The first evidence for the involvement of miRNAs in cancer came from Calin et al., who investigated the role of miR-15 and miR-16 in chronic lymphocytic leukemia [209]. Exosomes derived from biopsy specimens and serum samples serve as biomarkers for ovarian and lung cancer diagnosis [210,211]. Circulating miR-141 and miR-375 have been shown to be valuable biomarkers for prostate cancer diagnosis [212,213]. Exosomes isolated from human saliva and amniotic fluid contain hundreds of stable mRNAs and are valuable tools in pancreatic cancer and prenatal diagnostics [214,215]. An increased level of exo-miR21 in squamous cell carcinoma patients indicates tumor severity, progression, and aggressiveness [216]. Increased levels of miR-192 in serum exosomes of heart disease patients indicate the development of heart failure after acute myocardial infarction [217]. Exosomes enriched with miR-24-3p, miR-891a, miR106a-5p, miR-20a-5p, and miR-1908 affect cell proliferation and differentiation by down-regulating the MARK1 signaling pathway in nasopharyngeal carcinoma [190], whereas exosome-derived miR-302b suppresses cell proliferation via the TGFβRII/ERK pathway in lung cancer [218]. Exosomes derived from lung carcinoma showed different levels of the miRNAs, miR-378a, miR-379, miR-139-5p, and miR-200-5p compared to exosomes from normal tissue [219]. Therefore, circulating exosomal miRNAs may be useful for lung adenocarcinoma screening. Recent studies have suggested that the levels of exosomal miR-21-5p, miR-574-3p, and miR-141-5p are significantly upregulated in the urine of prostate cancer patients [220]. These data indicate that exosomal miRNAs may provide a new class of biomarkers for the early and minimally invasive diagnosis of cancer and inflammatory diseases.

### 6.6. Exosomes and Receptor Mediated Endocytosis

Exosomes are nanoscale membrane vesicles secreted from various types of cells. Exosomes mediate many physiological and pathological functions through carrying a variety of functional molecules and transfer information between the cells [59]. With the increasing use of exosomes in various biological functions, the role of exosomes in receptor-mediated endocytosis remains to be explored, and essential aspects of exosome function, such as the uptake mechanisms, are still unknown. There are multiple pathways that can mediate endocytosis, including phagocytosis, macropinocytosis, clathrin-mediated endocytosis, caveolae-mediated endocytosis, and clathrin- and caveolae-independent endocytosis [221]. Grapp et al. [222] demonstrated the importance of the folate receptor-α-mediated folate transport in the cerebrospinal fluid using folate receptor-α-positive and -negative exosomes in the brain parenchyma. Exosomes derived from erythroleukemia cells were internalized through phagocytosis or macropinocytosis [223]. Glioblastoma cell-derived exosomes were internalized through lipid raft-mediated endocytosis negatively regulated by caveolin-1 [224]. PC12 cell-derived exosomes deliver miR21 through clathrin-mediated endocytosis and macropinocytosis [225]. Nakase et al. [226] demonstrated the active induction of macropinocytosis by the stimulation of cancer-related receptors and significant enhancement cellular uptake of exosomes by oncogenic K-Ras-expressing MIA PaCa-2 cells. Huang et al. [227] reported that exosomal transfer of vasorin promotes the migration of human umbilical vein endothelial cells via receptor-mediated endocytosis of exosomes. Mesenchyma stem cell-derived exosomes promote axonal growth and can deliver their selective cargo miRNAs into and activate their target signals in recipient neurons via internalization mediated by the SNARE complex [228]. Several studies suggest that exosomes can be internalized by way of fusion and/or endocytosis [229,230]. Dendritic cell-derived exosomes bind to the plasma membrane, delivering their contents through the fusion or hemi-fusion of the two membranes [231]. Recently, Banizs et al. [232] observed that the uptake of endothelial-derived exosomes into endothelial cells was largely an energy-dependent process using a predominantly receptor-mediated, clathrin-dependent pathway. Therefore, exosomes may utilize several different mechanisms of uptake in the same cell and at different times. However, the receptor-mediated endocytosis plays a critical role in the entry of exosomes into the cells. 

### 6.7. Role of Exosomes in Cell Proliferation and Differentiation

Cell–cell communication is an essential factor for ligand concentration, receptor expression, and the integration of diverse signaling pathways. Particularly, secreted exosomes play a critical role in cell-cell communication mediators in physiological and pathological scenarios [233]. Exosomes exhibited a significant role in tumor growth and invasion in tumor-associated angiogenesis. Platelet-derived exosomes have been shown to induce tumor chemotaxis, proliferation, invasion, and expression of angiogenic factors and also promote thrombus formation and metastasis [234,235]. Proliferation of cancer cells is an essential event in cancer. Previous studies reported that tumor-derived exosomes can induce tumor cell proliferation. Glioblastomaderived exosomes induced cell proliferation of the human glioma U87 cell line [201]. For example, an autocrine induction of cellular proliferation was observed in chronic myeloid leukemia [236]) and in human gastric cancer [237,238] via phosphatidylinositol 3-kinase/protein kinase B (PI3K/AKT) and MAPK/ERK signaling pathways [237]. T24 tumor cell-derived exosomes induce cell proliferation in human bladder cancer cells T24 and 5637 through activation of the Akt and ERK pathways [239]. A recent study reported the very interesting observation of the promotion of in vivo growth of murine melanomas by systemic treatment of mice with melanoma-derived exosomes, which accelerated growth and inhibited apoptosis of melanoma tumors in vivo [240]. Exosomes derived from hypoxic prostate cancer cells induced increased invasiveness and motility of naïve human prostate cancer cells [241]. Exosomes containing the transcript of the enzyme Telomerase hTERT mRNA contributed to the establishment of pre-metastatic niches by increasing fibroblast proliferation and the lifespan [242]. Exosomes from senescent fibroblasts induce the proliferation of MCF-7 human breast cancer cells [243]. Cancer-Associated Fibroblasts (CAFs) derived exosomes are also capable of supporting tumor growth by providing nutrients to malignant cells [244]. Wnt4-enriched exosomes derived from hypoxic colorectal cancer cells promoted Beta-Catenin nucleartranslocation and proliferation of ECs [245]. Gastric cancer-derived exosomes induced NF-kB activation in macrophages, leading to an increase in the expression of pro-inflammatory factors such as IL-6 and TNF-a, in turn promoting the proliferation of gastric cancer cells [186].

Exosomes play a critical role in several biological processes, including intercellular communication, immune function, and the development and differentiation of stem cells [246]. For instance, mesenchyma stem cells have the potential to generate many different cell types that could replace lost cells in injured or dead tissues by the process of differentiation. Recently, several studeis have been dedicated to exploring the use of exosomes as inducers for differentiation in a variety of stem cells. A recent study suggests that proosteogenic exosomes isolated from cell cultures induce the lineage specific differentiation of naïve MSCs in vitro and in vivo [247]. miR-223 molecules carrying macrophage-derived microvesicles were transported to a variety of cells, including mono cytes, endothelial cells, epithelial cells, and fibroblasts. The results found that microvesicles induce the differentiation of macrophages [248]. EVs of endothelial origin, released from ischemic muscle, induce the differentiation of bone marrow-derived progenitor cells (BM-MNCs) into endothelial cells both in vitro and in vivo and subsequently promote postnatal vasculogenesis [106]. Exosomes secreted from monocytes stimulate the osteogenic differentiation of MSCs [249]. Exosomes derived from Burkitt’s lymphoma cell lines induce proliferation and differentiation [250]. Osteoblast-derived lysosomal membrane protein 1 (LAMP1) positive exosomes carry the RANK ligand, osteoprotegerin (OPG), and TRAP 186 enzymes, increasing osteoclastogenesis [251]. Osteoclast precursor-derived exosomes stimulate the differentiation ability of osteoclasts into mature phenotypes [252]. Mineralizing osteoblasts produced by exosomal miRNAs promote the osteogenic differentiation of ST2 cells. Osteo-miRNAs regulate osteoblast differentiation and function through Wnt signaling, insulin signaling, TGF-b signaling, and calcium signaling [253]. Mesenchymal stem cell-derived exosomes show a potential effect on the alteration of microRNA profiles and induce osteogenic differentiation, depending on the stage of differentiation, through modulation of the Wnt signalling pathway and endocytosis [254]. Hematopoietic stem cell-derived exosomes promote the hematopoietic diiferentiation of mouse embryonic stem cells through inhibiting the mir126/Notch pathway [255]. Exosomes derived from bladder cancer cells are internalized by fibroblasts and promote the proliferation and expression of CAF markers. This study found that exosomes are novel modulators of stromal cell differentiation [256]. 

Tenocytes-derived exosomes induce the tenogenic differentiation of MSCs in a modulation TGF-β signalling dependent manner [257]. Exosomes-mediated differentiation plays a significant role in treating tissue damage and also provides suitable cell resources to create tissue-engineered grafts for tissue repair.

## 7. Therapeutic Applications of Exosomes

The remarkable therapeutic potential of exosomes was not anticipated when they were initially discovered in 1983. Today, exosomes are seen as having enormous potential in biomedical applications. Exosomes participate in intercellular communication by delivering their contents, such as functional proteins, mRNA transcripts, and miRNAs, to recipient cells, with or without direct contact between cells. Thereby, they influence physiological and pathological processes. Due to the salient features of exosomes, they are able to reduce inflammation, cross the blood-brain barrier, enable multiple intravenous (IV) dosing without any side effects, and enhance neural and motor function. Exosomes play a significant role in prognosis and diagnosis in a wide range of pathological conditions, such as cancer, neurodegenerative disorders, liver and kidney disease, and numerous cardiopulmonary disorders (Figure 7). Recent studies have demonstrated that exosomes are novel therapeutic reagents [258]. Exosomes derived from MSCs have been tested in models of various diseases, such as respiratory, cardiovascular, neurological, musculoskeletal, hepatic, gastrointestinal, dermatological, and renal disease [258]. MSC-derived exosomes inhibit the expression of pro-inflammatory cytokines to exert anti-inflammatory effects and promote tissue regeneration by enhancing extracellular matrix remodeling [193,259,260,261,262,263]. Similarly, exosomes secreted from induced pluripotent stem cells, embryonic stem cells, and cardiac progenitor cells have therapeutic effects similar to MSC-derived exosomes [264,265,266]. MSC-derived EVs recapitulate the immunomodulatory and cytoprotective activities of their parent cells [267]. Bovine milk-derived exosomes attenuate arthritis [268]. DC-derived exosomes incubated with cancer antigens induced a cancer-specific T cell response [269]. Exosomes derived from B16BL6 murine melanoma cells containing melanoma antigens induce B16BL6-specific T-cell responses and inhibit tumor growth [270,271]. Bone marrow MSC-derived exosomes protect against various types of disease conditions, including myocardial ischemia/reperfusion injury [82], hypoxia-induced pulmonary hypertension [261], and brain injury [272]. Human umbilical cord MSC-derived EVs protect against acute renal injury and liver fibrosis [273,274]. Subcutaneous/intradermal injections of DEX lead to the stabilization of disease or an objective tumor response [275]. Tumor cell-derived exosomes stimulate immunosuppression through the promotion of T cell apoptosis, the suppression of dendritic cell differentiation and NK cytotoxicity, and the stimulation of immunosuppressive myeloid suppressor cells and regulatory T cells [276]. Higher concentrations of intracellular doxorubicin (DOX) and lower levels of cardiotoxicity are observed when this chemotherapeutic agent is loaded into exosomes compared to when it is administered systemically [277]. Similarly, increased neoplastic tropism and cytotoxicity is observed in MDR pulmonary metastases after treatment with macrophage EV-encapsulated paclitaxel [278]. Overexpression of miR-122 in adipose-derived mesenchymal stem cell exosomes enhances the inhibition of carcinoma growth and increases chemotherapy sensitivity in mouse xenograft models [279]. Similarly, marrow stromal cell-derived, miR-146b-enriched exosomes are able to silence EGFR and inhibit the proliferation of tumor cells in a rat model of glioma [280]. Furthermore, exosomes enriched with RAD51 and RAS52 siRNAs induce the death of fibrosarcoma cells, which indicates that EVs may be used as vectors in RNAi-based gene therapy [281].

EVs are not only used in cancer therapy, but have also been used to treat *diabetes mellitus*. For example, the intravenous injection of exosomes derived from human urinary stem cells reduces urinary albumin and podocyte apoptosis and increases the proliferation of glomerular endothelial cells in streptozotocin-treated rats. This study demonstrates that EVs may be a novel approach for the treatment of diabetic nephropathy [282]. Bone marrow stem cell-derived exosomes show neurorestorative effects, including a reduction in blood-brain barrier leakage and hemorrhage and an increase in axon and myelin density in rats with type II diabetes [283]. Mesenchymal stem cell-derived exosomes improve cognition by repairing oxidative damage in neurons and astrocytes in cognitively impaired diabetic animals [284]. Exosomes obtained from the conditioned media of MSCs reduce infarct size in a myocardial ischemia/reperfusion (I/R) model [82]. Arslan et al. [285] demonstrated a similar reduction in infarct size in a model of I/R following the administration of MSC-derived exosomes, 5 min prior to reperfusion. GATA-4-overexpressing MSC-derived exosomes increase cardiomyocyte survival and reduce cardiomyocyte apoptosis [263]. Cardiac progenitor cell-derived exosomes enriched in miR-451/144 promote cardioprotection by increasing cardiomyocyte survival in vivo, in a model of I/R and in vitro, in H9c2 cells [286]. Exosomes loaded with the chemotherapeutic agents, paclitaxel, doxorubicin, and withaferin by passive loading, release these agents slowly and effectively inhibit the proliferation of A549 lung cancer cells in vitro and tumor cells in vivo [287]. Moreover, when these drugs are administered within exosomes, they show lower IC50 values than when they are administered as free drugs. Curcumin-loaded exosomes isolated from the EL-4 mouse lymphoma cell line show a high stability and significantly suppress pro-inflammatory cytokines, such as IL-6 and TNF-α [288].

## 8. Technical and Biological Challenges of Using Exosomes

EVs have shown immense promise in therapeutic applications, due to their capacity for intercellular communication and the types of cargo molecules that they can carry. Despite this, the use of exosomes poses several technical and biological challenges. The first and foremost technical challenge is the production of sufficient quantities of exosomes [289]. In order to do achieve an effective dose response, at least 10–100 μg of exosomes is required. However, the yield from 1 mL of culture medium is usually less than 1 µg [290,291]. In addition, exosomes isolated from biological fluids are highly variable, impure, and low in yield. Therefore, standardized pre-analytical steps are crucial to minimize artefacts in EV analysis. Other factors, including cell culture matrices and plastics, the culture medium composition and volume, cell passage, cell confluency and viability, and mycoplasma and other microbial contamination status, are important considerations when optimizing methods for exosome production and isolation [292]. The large-scale production of exosomes can be achieved up to 5–10 fold in a bioreactor [293]. Studies suggest that physical, chemical, and biological stress can induce the production of exosomes; however, the contents, therapeutic efficacy, and safety of such exosomes need to be thoroughly evaluated, due to possible contamination with apoptotic bodies [294]. Enhancing the production and purity of exosomes can be achieved by modifying conventional methods, such as ultracentrifugation, SEC, ultrafiltration, aqueous two-phase systems, immunoaffinity, polymer-based precipitation, and microfluidic devices, or by developing new technologies. The selection of culture media is an important consideration when developing an exosome production method. For example, exosomes isolated from culture media containing serum have several impurities, whereas serum-free conditions cause major stress to the cells and lead to altered EV secretion [75].

Although current techniques allow the production of sufficient quantities of exosomes, it is critical that they are uniform and of a high quality [289]. The purity and physicochemical properties of exosomes depend on the method of isolation. For example, conventional methods like ultracentrifugation and precipitation are common methods, but they are not suitable for therapeutic applications, due to the varying composition, size, subpopulations, aggregations, and protein coronas of exosomes isolated using these methods [75,84,295,296,297,298]. In addition, the appropriate storage of isolated exosomes is crucial for biological activity. Exosomes can be suspended in phosphate-buffered saline and stored at −80 °C. The further addition of trehalose can protect exosomes from cryodamage [75,299]. All these factors may influence the uptake and clearance of exosomes in cellular systems. Therefore, optimization is necessary to isolate pure exosomes, while maintaining their original composition, so that undesired side effects induced by contaminants or specific subpopulations are reduced and their therapeutic efficacy is enhanced. The therapeutic potential of exosomes can be improved by modifying them with functional molecules or the overexpression of miRNAs. For example, the modification of B16BL6-derived exosomes with CpG DNA decreased the required dose to induce cancer antigen-specific immune responses by 10–100 fold compared with unmodified exosomes [270]. Furthermore, the alteration of exosome composition by hypoxia treatment enhances the biological effects of the exosomes [300,301]. 

## 9. Conclusions and Future Perspectives

EVs have major roles under various normal and pathological conditions. Exosomes play a significant role in communication between tumors and their target cells. The release of exosomes is essential for events such as gene transfer and the targeting of therapeutic agents to specific cells. The release, composition, and content of exosomes depend on the state of the cells from which they originate. Although several new technologies are being developed, existing methods for exosome isolation and characterization are frequently used for prognostic and diagnostic purposes. Here, we have summarized the history, biogenesis, isolation, characterization, biological functions, biomarkers, and significance of exosomes and their potential role in therapeutic applications for various diseases. Exosome biogenesis utilizes several cellular machineries involving various proteins and lipids, depending on the cell type and cellular homeostasis. We have also discussed the principles, advantages, and disadvantages of various exosome isolation and detection techniques.

DLS and NTA methods are capable of analyzing exosome parameters such as size and number. Electron microscopy is the best technique to analyze the structural features of exosomes; however, an ideal method would be able analyze structural and biological characteristics using a single instrument. Therefore, more sophisticated techniques are necessary for the isolation and characterization of exosomes. However, the selection of appropriate methods for exosome isolation and characterization is necessary to improve the quality of the exosomes isolated and the validity of the results of their use. The major technical challenge in exosome detection in therapeutic applications is the ability to differentiate exosomes derived from normal cells from those derived from cells with pathological conditions. Another challenge is the inherent heterogeneity of exosomes. It may be necessary to design a combination of various quantification techniques to differentiate exosome subtypes in heterogeneous samples. This will open new avenues for exosome detection and quantification. One important feature of new methods should be the ability to isolate different subpopulations of vesicles, to which an origin and function can be attributed. Once such limitations are overcome, new methods to manipulate the biogenesis, isolation, composition, secretion, and interaction of exosomes will help increase our understanding of their function and will facilitate the development of novel therapeutic strategies.

## Figures and Tables

**Figure 1 cells-08-00307-f001:**
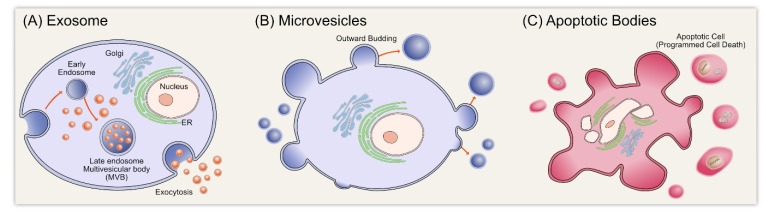
Schematic representation of extracellular vesicles.

**Figure 2 cells-08-00307-f002:**
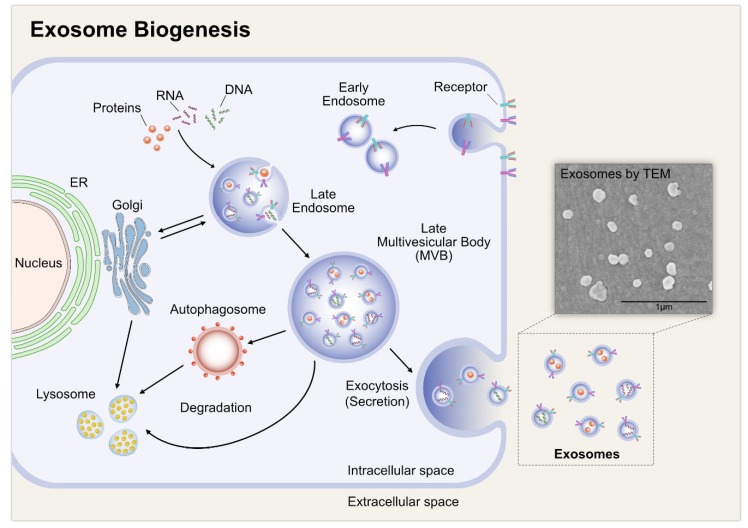
Biogenesis of exosomes. The insert shows biogenesis of exosomes of SHSY5Y cells.

**Figure 3 cells-08-00307-f003:**
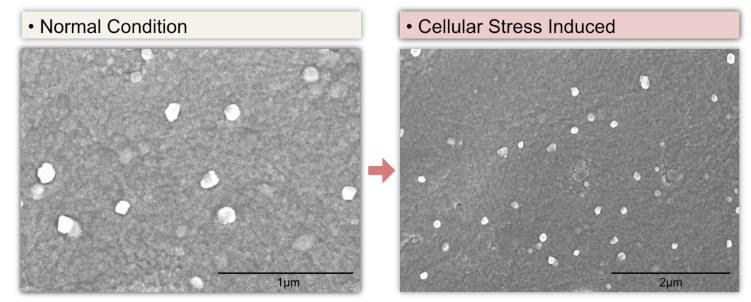
Silver nanoparticles induce biogenesis and secretion of exosomes.

**Figure 4 cells-08-00307-f004:**
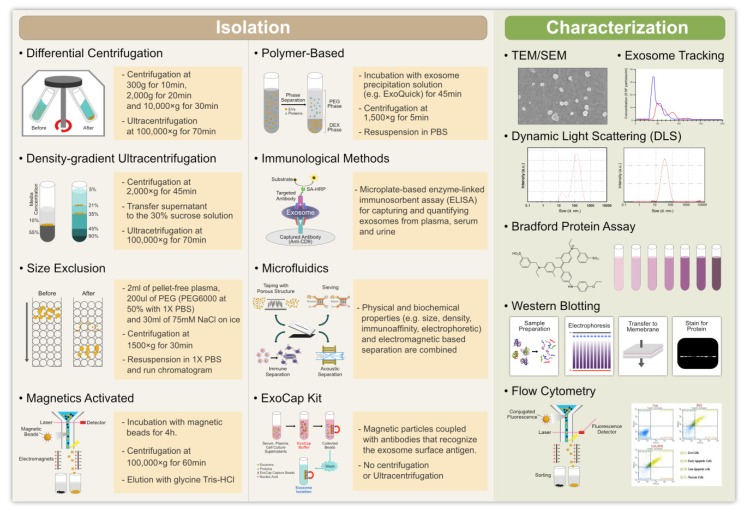
Various techniques used for isolation, characterization, and analysis of functional properties of exosomes.

**Figure 5 cells-08-00307-f005:**
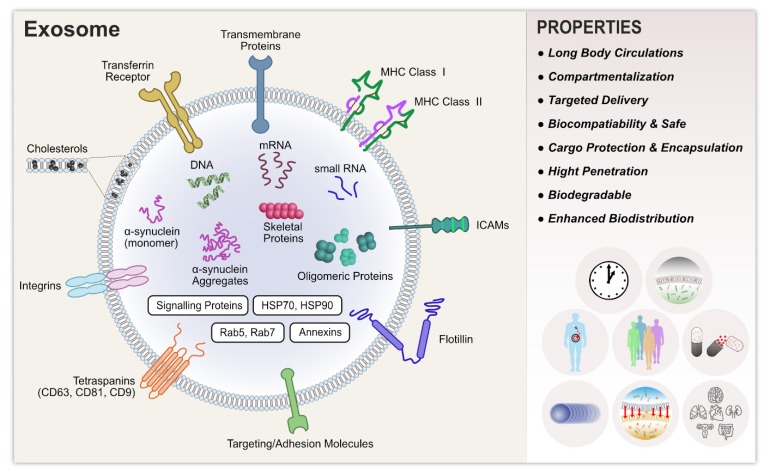
Typical structure of exosomes, properties, and functional attribution of various biomolecules present in exosomes.

**Figure 6 cells-08-00307-f006:**
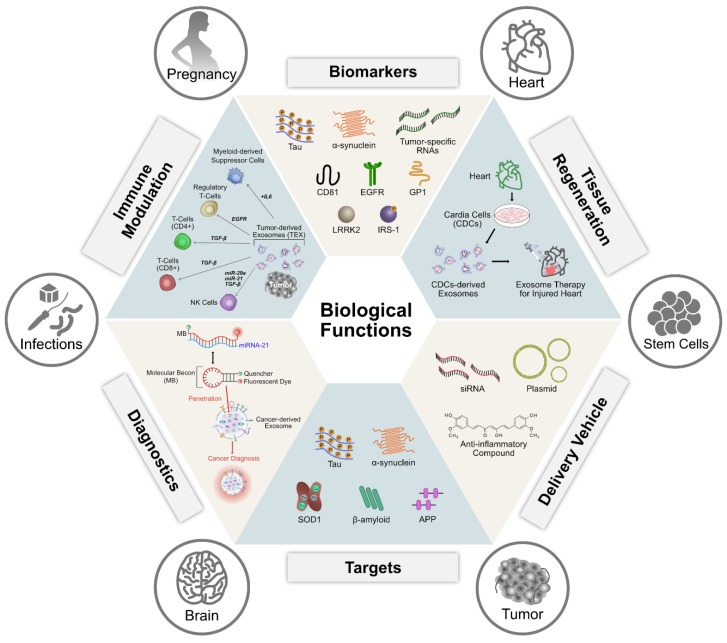
Biological function of exosomes.

**Figure 7 cells-08-00307-f007:**
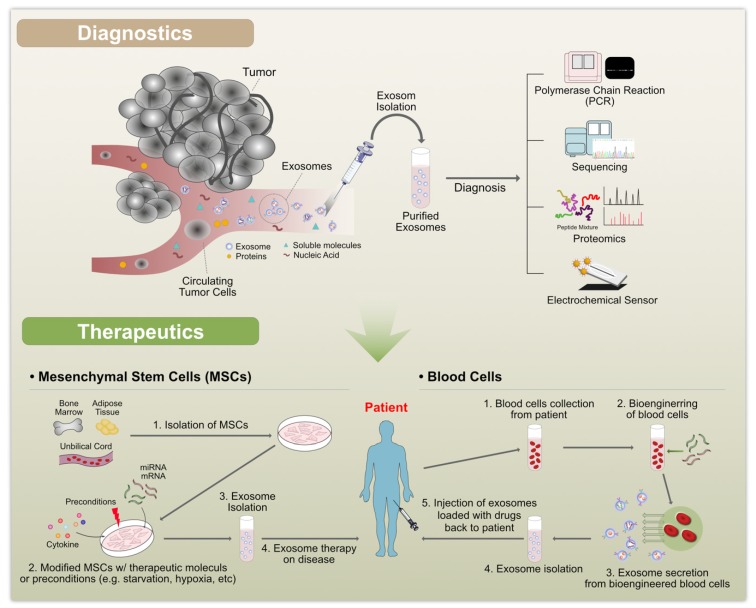
Diagnostic and therapeutic applications of exosomes.

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
