# Peer review of "Review of the Isolation, Characterization, Biological Function, and Multifarious Therapeutic Approaches of Exosomes"

_cells, 2019, doi:10.3390/cells8040307_

Round 1

Reviewer 1 Report

As a review paper it is expected that the authors will review the literature from the field which is shown through the citation of these primary manuscripts.  However, in this review a lot if not many of the citations review papers so it feels like most of what is being said has already been said.

In lines 70 to 71 the authors state that "the functions of EVs depend on their ability to interact with recipient cells and to deliver their contents of proteins, lipid and RNAs to these cells.  The specificity of binding to the target cells is governed by adhesion molecules, such as integrins."  I hope these authors will take us down the rabbit hole later in the manuscript with regard to this specificity as this in incredibly important to understand and is just now becoming evident.

There are multiple redundancies throughout this paper that extend it but also provide little such as describing exosomes as platelet dust in line 88 and 89 then again in line 100.

Organization within the manuscript could be better as there is between lines 100 and 110, which is in the "history" section of the paper a section that could be placed within the biogenesis section of the paper.  Perhaps even start the Biogenesis #3 section at line 100.  Remove all of line 100 first and start the paragraph..."EVs are classified into the following...

On page 4 and line 137/138 the authors describe the inhibition of ceramide reducing the release of of exosomal PLP.   Is the author trying to simply emphasize ceramide's control of exosomal release or are they trying to tell us something about PLP.  This occurs a number of times throughout this manuscript.  The author's then go on to say that exosome biogenesis can occur by either ESCRT- dependent or independent mechanisms and I do not believe that they presented enough for this to be the conclusion of this section of the review.  Seems out of place.

I am not sure what lines 152 through 156 have to do with biogenesis as it is stating the names of a number of proteins that are contained in exosomes and calling them biomarkers.  I would think it important to describe why these are considered biomarkers rather than simply calling them such without any description and they are not described as having a role in biogenesis.

Line 167 should be checked as it says "...but it is not suitable clinical samples..." and should read "...but is is not suitable for exosome isolation from clinical samples..."

A beneficial addition to this paper would be to build a table of the commercially available products for exosome purification and to provide judgement as to which would benefit the researcher and which would not.  Also identify price breakdown for purpose of making selection easier.  For instance, line 218 introduces PEG.  PEG is the primary ingredient in the EXOQuick products from SBI.

An important paper that details NTA which should be added to the section on Nanoparticle tracking analysis and NTA is by Rosalia de Necochea-Campion et al., 2018 in the journal Biomedical Physics and Engineering Express.  This manuscript provides a practical approach for establishing an EV characterization protocol for anyone wanting to compare EV content among similar biological samples.

A table comparing side by side the exosome characterization with the machines required and the approximate cost for the technology would be helpful, along with pros and cons between DLS, NTV, TRPS, AFM etc.

In certain areas of the paper the authors say exosomes carry proteins, lipids, DNA and RNA and then in others they say tRNA, proteins enzymes and lipids and leave out RNA or DNA or even more specialized with microRNA as in other areas of the paper.

The author's use exosomes, MV's and vesicles somewhat interchangeably throughout this manuscript and should try to keep it simple, especially in a single section.

An emerging area in the exosome field is receptor - mediated uptake.  There is little in this extensive review on how exosomes are recognized and allowed to deliver their payloads to the recipient cells.  Adding a section on uptake would round out this review.

Why is there not a separate section for exosomes and cell proliferation and differentiation as there is for angiogenesis?  In the angiogenesis section and in the exosomes as biomarkers, exosome content is discussed briefly as to their role in cell proliferation and differentiation and this should be discussed in its own section.

It is not clear why Biogenesis of exosomes (section 3) and factors influencing biogenesis (section 8 are not together.  They could even be combined into one section.

Author Response

Response to the reviewer comments-1

Comments and Suggestions for Authors

We immensely thank the reviewer and the editor for their valuable and constructive comments that greatly facilitated us for improving the overall quality of the manuscript. As per the editor’s and reviewers’ constructive comments, the corrections were carried out in the manuscript. We hopefully believe that we have addressed all the comments mentioned by the reviewers carefully and precisely. All the changes are highlighted in yellow color in the revised manuscript. In addition, this manuscript was proof read by native English speaker by Editage editing company, Seoul, South Korea.

As a review paper it is expected that the authors will review the literature from the field which is shown through the citation of these primary manuscripts.  However, in this review a lot if not many of the citations review papers so it feels like most of what is being said has already been said.

First of all thanks to the reviewer for excellent observation, constructive comments and encouragement to improve overall quality of the manuscript. We immensely thank the editor for their valuable and constructive comments. As per the reviewrs’s constructive comments, we have gone through the entire manuscript. We absolutely agree with reviewer, although we have cited review article, mostly we cited primary manuscripts rather than reviews. For your kind information we have cited only 18% of review articles and 82% of primary manuscripts. We followed nature review articles as model system, usually all nature review articles cited 20% of other review articles in a single review. In addition, according to the reviewer suggestions we included two new topics such as receptor mediated endocytosis and role of exosomes on cell proliferation and differentiation, these two topics were also cited only primary manuscripts.

In lines 70 to 71 the authors state that "the functions of EVs depend on their ability to interact with recipient cells and to deliver their contents of proteins, lipid and RNAs to these cells.  The specificity of binding to the target cells is governed by adhesion molecules, such as integrins."  I hope these authors will take us down the rabbit hole later in the manuscript with regard to this specificity as this in incredibly important to understand and is just now becoming evident.

Thanks to the reviewer for critical observation. According to the reviewer comments, the line 70 and 71 moved to respective section.

There are multiple redundancies throughout this paper that extend it but also provide little such as describing exosomes as platelet dust in line 88 and 89 then again in line 100.

Thanks to the reviewer for thought-provoking comments. According to the reviewer comments we deleted the repetitive word exosomes as platelet dust.

Organization within the manuscript could be better as there is between lines 100 and 110, which is in the "history" section of the paper a section that could be placed within the biogenesis section of the paper.  Perhaps even start the Biogenesis #3 section at line 100.  Remove all of line 100 first and start the paragraph..."EVs are classified into the following...

Thanks to the reviewer for excellent suggestion. According to the reviewer comments we moved the line from 100 to 110 of history section into biogenesis, now made one section as biogenesis and we merged both history and biogenesis as one section.

On page 4 and line 137/138 the authors describe the inhibition of ceramide reducing the release of of exosomal PLP.   Is the author trying to simply emphasize ceramide's control of exosomal release or are they trying to tell us something about PLP.  This occurs a number of times throughout this manuscript.  The author's then go on to say that exosome biogenesis can occur by either ESCRT- dependent or independent mechanisms and I do not believe that they presented enough for this to be the conclusion of this section of the review.  Seems out of place.

Response to your first question, we emphasized the importance of ceramide involvement in biogenesis rather than PLP. Then we checked the word ceramide, now it is found that necessary places.

Response to your second question, we deleted the sentences exosome biogenesis can occur by either ESCRT- dependent or independent mechanisms due to lack of evidences.

I am not sure what lines 152 through 156 have to do with biogenesis as it is stating the names of a number of proteins that are contained in exosomes and calling them biomarkers.  I would think it important to describe why these are considered biomarkers rather than simply calling them such without any description and they are not described as having a role in biogenesis.

Thanks to the reviewer for critical observation. We moved the sentences from 152 to 156 to biomarker section rather than included in biomarkers section.

Line 167 should be checked as it says "...but it is not suitable clinical samples..." and should read "...but is is not suitable for exosome isolation from clinical samples..."

We apologize for the error. According to the reviewer comments we rectified the sentences in the revised manuscript. 

A beneficial addition to this paper would be to build a table of the commercially available products for exosome purification and to provide judgement as to which would benefit the researcher and which would not.  Also identify price breakdown for purpose of making selection easier.  For instance, line 218 introduces PEG.  PEG is the primary ingredient in the EXOQuick products from SBI.

Really, we appreciate the wonderful idea of reviewer. While writing this review, we have gone through several papers, among several papers particulars review paper has been focused only on methods related to commercially available products for exosome purification and advantages and disadvantages.  Therefore we didn’t include and repeat the same.

An important paper that details NTA which should be added to the section on Nanoparticle tracking analysis and NTA is by Rosalia de Necochea-Campion et al., 2018 in the journal Biomedical Physics and Engineering Express.  This manuscript provides a practical approach for establishing an EV characterization protocol for anyone wanting to compare EV content among similar biological samples.

According to the reviewer comments, we have cited Rosalia de Necochea-Campion et al., 2018 in the NTA section.

A table comparing side by side the exosome characterization with the machines required and the approximate cost for the technology would be helpful, along with pros and cons between DLS, NTV, TRPS, AFM etc.

Thanks to the reviewer for excellent idea. It can be mentioned the machines required for characterization of exosomes (this part has discussed in this review) and it is very difficult to comprehensive the approximate cost for the technology, every places the cost is different depends on company, availability of machines, labor and type of machines etc.  For your kind information, we have already mentioned about pros and cons about each methods in our review. I hope the reviewer could understand our difficult condition also.

In certain areas of the paper the authors say exosomes carry proteins, lipids, DNA and RNA and then in others they say tRNA, proteins enzymes and lipids and leave out RNA or DNA or even more specialized with microRNA as in other areas of the paper.

Thanks to the reviewer for clear analysis of writing. We rectified the error and we brought all the information into uniformity.

The author's use exosomes, MV's and vesicles somewhat interchangeably throughout this manuscript and should try to keep it simple, especially in a single section.

Thanks to the reviewer for vital question. As you know very well cells release into the extracellular environment diverse types of membrane vesicles of endosomal and plasma membrane origin called exosomes and microvesicles, respectively. Extracellular vesicles are classified on the basis of their cellular origin, biological function, size, and most commonly by their biogenesis. On the basis of their formative processes, there are three main classes: microvesicles, apoptotic bodies, and exosomes. Microvesicles originate from the plasma membrane as a result of outward budding and fission of membrane vesicles from the cell surface. Apoptotic bodies result from the blebbing of the plasma membrane during apoptosis . Exosomes, the focus of this article, derive from intracellular inward budding of the limiting membrane of endocytic compartments that form multivesicular bodies (MVB), which release these vesicles in the form of exosomes. We have used the word exosomes properly and some places we have used microvesicles if necessary. Furthermore, according to the reviewer comments, we gone through the entire manuscript and rectified the repetition of the words.

An emerging area in the exosome field is receptor - mediated uptake.  There is little in this extensive review on how exosomes are recognized and allowed to deliver their payloads to the recipient cells.  Adding a section on uptake would round out this review.

Thanks to the reviewer for kind note. According to the reviewer comments we included the role of exosomes in uptake and endocytic mechanism of exosomes as new section in 6.6

Why is there not a separate section for exosomes and cell proliferation and differentiation as there is for angiogenesis?  In the angiogenesis section and in the exosomes as biomarkers, exosome content is discussed briefly as to their role in cell proliferation and differentiation and this should be discussed in its own section.

Thanks to the for constructive comments. According to the reviewer we included the role of exosomes in cell proliferation and differentiation as new section 6.7 in the revised manuscript.

It is not clear why Biogenesis of exosomes (section 3) and factors influencing biogenesis (section 8 are not together.  They could even be combined into one section.

Thanks to the reviewer for excellent comments. Initially we would like to include both biogenesis and factors influencing biogenesis of exosomes in single section. During preparation of this review, we realized that if we include both biogenesis and factors influencing biogenesis of exosomes, the readers may not give more attention and also there is no importance of factors influencing biogenesis, therefore we placed different places to emphasize the importance of factors influencing biogenesis of exosomes and also so far there is no review has discussed the factors influencing biogenesis of exosomes particularly nanoparticles. However, according to the reviewer comments we shifted the factors influencing of biogenesis from bottom to next to biogenesis section.

Once again thanks to the reviewers for wonderful comments to improve the overall quality of the manuscript.

Reviewer 2 Report

The manuscript (Review of the isolation, characterisation, biological function, and multifarious therapeutic approaches of exosomes) by Gurunathan et al, is a very well written and comprehensive review of the history and current state of the art of exosomes.

This manuscript is an excellent account of exosomes, their biological functions and potential applications that will be of benefit to the field.

I have noted some minor typographical errors and suggestions below.

The manuscript uses too many abbreviations. Although abbreviations are tempting and often necessary to improve reading flow. Given the large number used within this manuscript, I would suggest that avoiding unnecessary abbreviations may improve the reading flow and widen the appeal of the manuscript beyond experts in the field. e.g. endothelial cells (ECs) is arguably not really needed as the abbreviations doesn't improve reading flow significantly. My suggestion would be that the authors remove any abbreviations that aren't required and use the full words.

Line 107: typo in the sentence

Line 123/124: font inconsistency

Line 167/168: sentence structure needs revision. This is essentially a list so should use commas and fewer 'and' words

Line 174-183: this paragraph needs reorganising to mention analytical and preparative ultracentrifugation before the two examples of preparative ultracentrifugation. Is 1000,000 x g a typo? Fig.3 only suggests ultracentrifugation speeds up to 100,000 x g

Figure 3: the figure contains some text formatting errors

Line 347: period after morphology needs deleting

Line 351: Prior to the examination under TEM could be improved, perhaps Prior to examination by TEM.

Line 405: period after cells needs deleting

Line 405: extra space after found 

Line 408-410: font inconsistency.

Line 439: font inconsistency.

Line 440: font inconsistency.

Line 590/591: duplication of associated with

Line 606: delete it

Line 634-637: font inconsistency.

Line 669: correct to diabetes mellitus

Line 707: font inconsistency.

Line 716: delete do

Author Response

Response to the reviewer comments -2

We immensely thank the reviewer for their valuable and constructive comments that greatly facilitated us for improving the overall quality of the manuscript. As per the editor’s and reviewers’ constructive comments, the corrections were carried out in the manuscript. We hopefully believe that we have addressed all the comments mentioned by the reviewers carefully and precisely. All the changes are highlighted in yellow color in the revised manuscript. In addition, this manuscript was proof read by native English speaker by Editage editing company, Seoul, South Korea.

Comments and Suggestions for Authors

The manuscript (Review of the isolation, characterisation, biological function, and multifarious therapeutic approaches of exosomes) by Gurunathan et al, is a very well written and comprehensive review of the history and current state of the art of exosomes. This manuscript is an excellent account of exosomes, their biological functions and potential applications that will be of benefit to the field. I have noted some minor typographical errors and suggestions below.

First of all thanks to the reviewer for excellent observation, constructive comments, positive response about our manuscript and excellent comments to improve overall quality of the manuscript.

The manuscript uses too many abbreviations. Although abbreviations are tempting and often necessary to improve reading flow. Given the large number used within this manuscript, I would suggest that avoiding unnecessary abbreviations may improve the reading flow and widen the appeal of the manuscript beyond experts in the field. e.g. endothelial cells (ECs) is arguably not really needed as the abbreviations doesn't improve reading flow significantly. My suggestion would be that the authors remove any abbreviations that aren't required and use the full words.

We absolutely agree with reviewer. We deleted unnecessary abbreviations in the revised manuscript.

Line 107: typo in the sentence

Thanks to the reviewer for critical observation of each and every word and sentences. According to the reviewer, we rectified the all the errors.

For your kind and additional information, due to other reviewer comments we have included additional content in the manuscript as extra lines or sentences or modified the entire line or paragraph, therefore the sentence or line indicated by you by line number may not be in the same position, but it will be slightly different place with different line number.

Line 123/124: font inconsistency

Thanks to the reviewer, we brought same font.

Line 167/168: sentence structure needs revision. This is essentially a list so should use commas and fewer 'and' words

According to the reviewer we deleted repeated appearance of words in the sentences.

Line 174-183: this paragraph needs reorganising to mention analytical and preparative ultracentrifugation before the two examples of preparative ultracentrifugation. Is 1000,000 x g a typo? Fig.3 only suggests ultracentrifugation speeds up to 100,000 x g

Thanks to the reviewer we rectified the error in the figure.

Figure 3: the figure contains some text formatting errors

According to the reviewer we fixed the text error.

Line 347: period after morphology needs deleting

According to the reviewer we deleted after morphology

Line 351: Prior to the examination under TEM could be improved, perhaps Prior to examination by TEM.

According to the reviewer we changed prior to the examination under TEM could be improved, perhaps prior to examination by TEM

Line 405: period after cells needs deleting

According to the reviewer we delete

Line 405: extra space after found 

According to the reviewer we rectified the space.

Line 408-410: font inconsistency.

According to the reviewer we rectified the font size

Line 439: font inconsistency.

According to the reviewer we rectified the font size

Line 440: font inconsistency.

According to the reviewer we rectified the font size

Line 590/591: duplication of associated with

According to the reviewer deleted the repetitive word

Line 606: delete it

According to the reviewer deleted the word it

Line 634-637: font inconsistency.

According to the reviewer we rectified the font size

Line 669: correct to diabetes mellitus

According to the reviewer we rectified the error

Line 707: font inconsistency.

According to the reviewer we rectified the font size

Line 716: delete do

According to the reviewer we deleted the word do

Once again thanks to the reviewers for wonderful comments to improve the overall quality of the manuscript.
